# On Urinary Bladder Cancer Diagnosis: Utilization of Deep Convolutional Generative Adversarial Networks for Data Augmentation

**DOI:** 10.3390/biology10030175

**Published:** 2021-02-26

**Authors:** Ivan Lorencin, Sandi Baressi Šegota, Nikola Anđelić, Vedran Mrzljak, Tomislav Ćabov, Josip Španjol, Zlatan Car

**Affiliations:** 1Faculty of Engineering, University of Rijeka, Vukovarska 58, 51000 Rijeka, Croatia; ilorencin@riteh.hr (I.L.); nandelic@riteh.hr (N.A.); vmrzljak@riteh.hr (V.M.); car@riteh.hr (Z.C.); 2Faculty of Dental Medicine, University of Rijeka, Krešimirova 40/4220/1, 51000 Rijeka, Croatia; tomislav.cabov@fdmri.uniri.hr; 3Faculty of Medicine, University of Rijeka, Ul. Braće Branchetta 20/1, 51000 Rijeka, Croatia; josip.spanjol@medri.uniri.hr; 4Clinical Hospital Center, 51000 Rijeka, Croatia

**Keywords:** AlexNet, data augmentation, deep convolutional generative adversarial networks, urinary bladder cancer, VGG16

## Abstract

**Simple Summary:**

One of the main challenges in the application of Machine Learning in medicine is data collection. Either due to ethical concerns or lack of patients, data may be scarce. In this paper Deep Convolutional Generative Adversarial Networks (DCGAN) have been applied for the purpose of data augmentation. Images of bladder mucosa are used in order to generate new images using DCGANs. Then, combination of original and generated images are used to train AlexNet and VGG16 architectures. The results show improvements in AUC score in some cases, or equal scores with apparent lowering of standard deviation across data folds during cross-validation; indicating networks trained with the addition of generated data have a lower sensitivity across the hyperparameter range.

**Abstract:**

Urinary bladder cancer is one of the most common urinary tract cancers. Standard diagnosis procedure can be invasive and time-consuming. For these reasons, procedure called optical biopsy is introduced. This procedure allows in-vivo evaluation of bladder mucosa without the need for biopsy. Although less invasive and faster, accuracy is often lower. For this reason, machine learning (ML) algorithms are used to increase its accuracy. The issue with ML algorithms is their sensitivity to the amount of input data. In medicine, collection can be time-consuming due to a potentially low number of patients. For these reasons, data augmentation is performed, usually through a series of geometric variations of original images. While such images improve classification performance, the number of new data points and the insight they provide is limited. These issues are a motivation for the application of novel augmentation methods. Authors demonstrate the use of Deep Convolutional Generative Adversarial Networks (DCGAN) for the generation of images. Augmented datasets used for training of commonly used Convolutional Neural Network-based (CNN) architectures (AlexNet and VGG-16) show a significcan performance increase for AlexNet, where AUCmicro reaches values up to 0.99. Average and median results of networks used in grid-search increases. These results point towards the conclusion that GAN-based augmentation has decreased the networks sensitivity to hyperparemeter change.

## 1. Introduction

Urinary bladder cancer, as one of the most common malignant diseases of the urinary tract, is a consequence of a mutation in the bladder’s mucosa cells which causes their uncontrolled growth. Such growth shows a high tendency to spread to the rest of mucosa, but also to other parts of the human body. Above stated facts, urinary bladder cancer is showing elevated recurrence rates ranging from 61% in the first year, to 78% in the first five years. Such recurrence rates are among the highest compared to other malignant diseases. For this reason, diagnosis, treatment and follow-up are extremely challenging [1]. There is a multiplicity of urinary bladder cancer types, among which the most common are:urothelial carcinoma [2,3],squamous cell carcinoma [4,5],adenocarcinoma [6,7],small cell carcinoma [8,9] andsarcoma [10,11].

Most urinary bladder cancers are urothelial carcinomas or transitional cell carcinomas (TCC). These carcinomas are, in their papillary form, characterized by low-grade metastatic potential [12]. On the other hand, high-grade cancer and carcinoma in-situ (CIS) are characterized by higher metastatic potential. Unlike TCC, CIS lesion of urinary bladder mucosa is dominantly flat, making it harder to differentiate from benign growths.

The dominant medical examination for the diagnosis of urinary bladder cancer is cystoscopy, an endoscopic method where a probe called a cystoscope is inserted into the urinary bladder via ureter. Modern cystoscopes are equipped with confocal laser endomicroscopes (CLE) that allow in-vivo evaluation of the mucosa, without the need for biopsy and patho-histological examination. Such a method called an optical biopsy, shows high results from the point of view of detecting papillary lesions. On the other hand, the same method has failed in the detection of CIS, and its accuracy does not exceed 75%. To increase the likelihood of a positive outcome, the goal is an accurate and timely diagnosis of CIS, which, as already mentioned, is characterized by its high metastatic potential [13,14].

With aim of increasing accuracy, machine learning (ML) algorithms are introduced and integrated with CLE-based cystoscopy [15]. Such integration can be of particular importance especially in the case of CIS recognition. There were several research works based on ML utilization for cystoscopic image classification. Research presented in [16] proposed the utilization of a multilayer perceptron (MLP) whose initial weights were pre-determined with a genetic algorithm. The authors of the research presented in [17] have proposed a Convolutional Neural Network-based (CNN) approach that achieved AUC of 0.98. CNN-based approach is also presented in [18], where authors have achieved accuracies up to 98%. A similar approach is presented in [19], where authors have proposed utilization of pre-defined networks in order to achieve high classification performances. The best results have been achieved with Xception architecture (F1=99.52%). A hybrid approach is presented in [20] where authors have proposed a combination of Laplacian edge detector and MLP that follows CNN methodology. By using this approach, AUC values up to 0.99 were reached. From the presented overview, it can be noticed that CNN-based architectures are playing a lead role in ML-based algorithms for the classification of cystoscopic images. However, regardless of the encouraging results in this area, there is always room for improvement, both from CNN and data preprocessing standpoint.

The collection of data for Artificial Intelligence (AI) studies in the field of medicine, especially in the classification of rarely appearing diseases—such as specific types of carcinoma—can be complex. Not only can the data collection process be hard due to a low number of patients on which the testing procedures need to be performed, ethical questions can arise. Not all patients may consent to the data collected during procedures performed with them as the subject being shared with non-medical researchers. Additionally, it can be hard to guarantee near equal numbers across various classes, as, naturally, the larger number of patients on who the testing procedure is performed are positive, due to the test usually being performed on symptomatic subjects. Due to this, there might be a larger number of positive test findings in the input, in comparison to the negative (healthy) patients [20,21].

Because of the above-listed issues, data augmentation is commonly used. Data augmentation refers to the process of generating new inputs for Artificial Neural Network (ANN) training. This process is commonly performed on datasets that consist of pictures as inputs. A common practice is the use of deterministic modifications on the images of the input dataset, where standard transformations are used [22,23]. These transformations include vertical and horizontal mirroring, rotating, and scaling of different image sections [24]. While these transformations can provide additional robustness to the model, they are still tightly connected to the existing data and may not provide needed variance to the input dataset.

The answer to this issue lies in the use of Generative Adversarial Networks (GAN) [25]. Through the use of GAN new images can be generated, which can then be used as inputs, and the resulting models can be tested using previously established metrics, to test the precision of models created using generated data.

The aim of this research is to examine the influence of GAN-based image data augmentation of CNNs for urinary bladder cancer diagnosis. This research is focused on increasing the performance of classifiers by applying dataset pre-processing, while using established CNN architectures that have achieved high classification performance in solving various problems in practice. From facts stated above, several questions could be asked:Is it possible to utilize GAN fo urinary bladder cancer image data augmentation,Are classifier performance higher if an augmented set is used andHow does the share of data generated in the training set affect the performance of the classifier?

## 2. Problem Description

The proposed method for urinary bladder cancer diagnosis is based on the utilization of CNNs in order to classify images obtained by CLE, as presented in Figure 1. The diagnosis system is trained by using urinary bladder mucosa images that are obtained with optical biopsy and evaluated by standard patho-histological examination. The idea behind the described diagnosis system is to train CNN by using the dataset of confirmed images that will be used to evaluate new images.

Images obtained by CLE are divided into four classes: Non-cancer tissue, high-grade carcinoma, low-grade carcinoma, and CIS. The representation of each class is given in Figure 2. One of the challenges in the design of ML-based classifiers is the acquisition of sufficiently large datasets. Such characteristic is particularly emphasized in medical diagnostic, where accurate classification is of significant importance [26]. For these reasons, the aim of this research is to utilize GAN in order to artificially increase the training dataset that will result in higher classification performances.

The representation of each class is presented in Figure 2. An example of healthy mucosa is represented in Figure 2a, where no signs of malignant changes of bladder mucosa can be found. In this case, it can be stated that the observed tissue represents a non-malignant disease or a completely healthy mucosa. Figure 2b. represents a high-grade bladder cancer. In this case, the cells and tissue are poorly differentiated and abnormal, without geometrical structure or pattern. This grade is characterized by high metastatic potential [12]. Figure 2c represents a sample of low-grade cancer. In this case, in difference with the case of high-grade cancer, cells are well differentiated. Figure 2d represents CIS. This class is characterized by its flat form that significantly differs from papillary carcinomas. Furthermore, as it is in the case of high-grade bladder cancer, CIS is characterized by high metastatic potential [12].

## 3. Use of DCGAN Networks in Generation of Medical Data

In this section, a brief overview of a GAN-based image generating process will be presented. The overview of the process is relatively simple. Dataset is split into two parts—training and testing set, as is the standard ANN training practice [27,28]. The testing part of the dataset is set aside, while the training set is fed into the two parts of GAN—discriminator and generator [29]. A discriminator is trained to distinguish real and fake images, while Generator is trained to generate images. The generator adjusts its’ parameters in an attempt of generating images that are realistic enough that the discriminator does not detect them as false. Once this process is completed, the generated images are mixed with the original training set, providing an augmented training set. This augmented training set is then used to train the CNN and created models which are evaluated on images of the testing set. It is important to note that the testing set does not contain images generated via the described process. If generated images were of low-quality due to improper settings of the GAN, scores could still be high if they were used in the evaluation. By avoiding using the generated images within the test set, if they are of low-quality CNN model will be trained using inappropriate data—where inappropriate data means generated images which are massively inconsistent in comparison to real data. If the model is trained with such data, and then evaluated using real data the scores will be low. In this way, the data testing process evaluates the CNN and GAN quality as well, as both ANNs will have to achieve good results for the scores to be acceptable. This process is shown in Figure 3.

The ANN setup used consists of two separate ANNs—generator and discriminator. The generator and discriminator used in this research are both deep convolutional neural networks, which makes GAN used in this research a Deep Convolutional Generative Adversarial Network (DCGAN) [30].

Convolution is a mathematical operation defined on two functions (*x* and *h*) which produces a third function that expresses how the function *h* changes the shape of function *x* [31]. If *n* and *k* refer to indices of the discreete signals, it can be defined with:(1)$$C\left(x,h\right)\left[n\right]=x*h=\sum_{k=-\infty }^{\infty }h\left[k\right]x\left[n-k\right].$$

Equation (Equation 1) defines the one-dimensional, discrete convolution. Considering the fact that the convolution is performed on images, a two-dimensional definition is necessary, given by [31,32]:(2)$$C\left(x,h\right)\left[n_{1},n_{2}\right]=\sum_{k_{1}=-\infty }^{\infty }\sum_{k_{2}=-\infty }^{\infty }h\left[k_{1},k_{2}\right]x\left[n_{1}-k_{1},n_{2}-k_{2}\right],$$
or:(3)$$C\left(x,h\right)\left[n_{1},n_{2}\right]=\sum_{k_{1}=-\infty }^{\infty }\sum_{k_{2}=-\infty }^{\infty }h\left[k_{1}\right]h\left[k_{2}\right]x\left[n_{1}-k_{1},n_{2}-k_{2}\right].$$

Convolutional neural networks work by applying convolution onto the input of the layer [28,33]. They filter the image, capturing the spatial and temporal dependencies contained within. The role of convolutional networks is transforming images into smaller, easier to process forms, without losing the important features [34]. The convolution layer is defined using four parameters [35]:number of filters—which define the dimensionality of the output,the size of kernel (*h*)—which specify the dimensions of convolution window,strides—which specify convolutional strides along the height and width, andpadding—which defines whether padding will be applied to the output of convolution or should the size remain the same.

Assuming the input image of dimensions 28×28×1 pixels, where the first number represents the height of the image in pixels, the second represents the width, and final is the number of color channels—in this case one, as a gray-scale image is assumed. The size of the image is defined, for both its height and width using the following [36]:(4)$$L^{\prime }=\frac{\left(L-F+2P\right)}{S}+1,$$
where:$L^{\prime }$ is the output dimension (width or height) of the image,*L* is the dimension (width or height) size of the image,*F* is the dimension of the kernel in the same direction (width or height) as the image dimension being transformed,*P* is the size of padding applied to the image, and*S* is the total number of strides.

The “same” padding is used. This padding will assure that the input image gets fully covered by the filter and specified stride. As this is the setting used, the Equation (Equation 4) can be simplified to [36]:(5)$$L^{\prime }=\frac{L}{S}.$$

Transposed convolution allows the input to be expanded instead of compressed as it is when using convolution. It is commonly used in upscaling, but its role in DCGAN is the generation of an image from randomized input data using learned weights stored within the kernel *h*. Transposed convolution is performed by taking the original image *x* and kernel *h*. Then zero-padding equal to the dimensions of *h*, minus 1, is added to the image *x*. The convolution is then performed and the calculation for each element in the new matrix C(x,h) is performed using Equation (Equation 3), and, to differentiate it, commonly marked using:(6)C(x,h)=h∗∗x.

The size of the resulting matrix is defined with [32,36]:(7)$$L^{\prime }=\left(L-1\right)*S+F-2P,$$
which can, when using “same” padding be simplified similar to Equation (Equation 5) [32,36]:(8)$$L^{\prime }=L\cdot S.$$

The networks in question are referred to as “adversarial”, as they are working “against each other” [37]. The discriminator is trained as a classifier—using the original collected data—with the goal of differentiating between real and fake images [38]. The generator is created to generate images, with the goal of tricking the discriminator [30,38]. It takes a randomized input, which is in this case a uniformly distributed random vector of 100 elements in a range between 0.0 and 1.0. This randomized input is fed into the generator which, using two-dimensional transposed convolution, generates the image. The generated image is then fed into the discriminator which classifies it as belonging to the original dataset or not. In other words; it discriminates between false (generated) images and real ones [38]. If the discriminator determines the image generated is fake, the parameters of the generator are adjusted, and the process repeats [39]. In this way, the generator continually adjusts its parameters, until the images produced by it are indistinguishable from the original dataset images to the discriminator [40].

The input to the generator shaped (1, 100) is fed into a densely connected input layer shaped ($w_{out}/4,h_{out}/4,256$) where $w_{out}$ represents the expected width of the generated image, and $h_{out}$ represents the expected height of the generated image. This input passes through batch normalization, which normalizes the output of the previous layer through the subtraction of the image batch mean and dividing by the image batch standard deviation. Such an approach provides higher stability to the network as it avoids too large of a value appearing in the layer output [41]. It also provides a more independent learning environment to the layers surrounding it [33,41]. The output is then passed through LeakyReLU activation function. LeakyReLU comes from Rectified Linear Unit activation function, which maps the positive input directly to the output, while eliminating the negative outputs—defined with F(x)=max(0,x). LeakyReLU functions in a similar manner, but instead of completely eliminating negative values it just lowers their value by the factor of 0.01 [42]. LeakyReLU can be defined using [43]:(9)$$F\left(x\right)=\left\{\begin{array}{c} x,x\in {I\;R}_{0}^{+}\\ 0.01\cdot x,x\in {I\;R}^{-} \end{array}\right.$$

The generator then uses two two-dimensional transposed convolution layers to further expand the image, going to shapes of ($w_{out}/4,h_{out}/4,128$) to ($w_{out}/2,h_{out}/2,64$) and finally to ($w_{out},h_{out},1$). The output of each transposed convolution layer, except the last, is processed through batch normalization and LeakyReLU activation layers. It can be seen, that the changes to the image dimensions follow Equation (Equation 8). For example, generating process of 28 × 28 image starts with the aforementioned (1, 100) uniformly randomized vector, which is mapped using a densely connected layer into the shape of (1, 12,544). Such a vector is reshaped into the three-dimensional layer shaped (7, 7, 256). Through the process of transposed convolution, this is transformed into (14, 14, 64), and finally (28, 28, 1). This form represents the desired output shape. The third dimension is a value defined via the arbitrary number of filters contained in the two-dimensional transposed convolution layer.

The discriminator is also defined as CNN. It consists of two two-dimensional convolution layers, followed by leakyReLU activation and a Dropout Layer. The dropout layer applies dropout regularization to the output of a previous layer. Regularization is important in avoiding the over-fitting of ANN to certain prominent features [27]. Dropout achieves this by setting a fraction of inputs to the layer to 0. In this case, the rates for dropout are 0.3 (30%) for both dropout layers. Finally, a flatten layer is used to flatten the vector of *n*-dimensional matrices into a vector which can be interpreted by the final layer of the CNN—a dense layer containing a single output neuron. The value of the neuron in the final layer defines the output of the discriminator. The architecture and process of training are shown in Figure 4. The transformation of image sizes within the network follows the Equation (Equation 5). For example, if the input image has the dimensions of (28, 28, 1)—which is the expected image size—the first two-dimensional convolution layer transforms this into (14, 14, 64)—where the first two dimensions follow the Equation (Equation 5), and the third being arbitrarily defined number of filters. The second convolutional layer transforms this into (7, 7, 128), by following the same rules. Finally, this is flattened using the flatten layer into a vector of dimensions (1, 6272), where the number of 6272 elements is obtained by multiplying all dimension in the output tensor of the previous layer (7×7×128=6272), in order for both of these layer outputs to have the same number of elements. Finally, this is transformed into a single output neuron using a densely connected layer with the same number of connections as the outputs of the flatten layer—6272. It is important to note that in GAN networks the output of the generator needs to fit not only the expected generated image size, but also the input to the discriminator. When working with DCGANs this means that convolution and transposed convolution layers need to be adjusted in such a manner that this can be achieved.

Loss *L* is a measure of difference between the expected and actual input of the neural network. In other words, it defines the error of the neural network. By lowering the value of loss, a more precise ANN is achieved, and this is the goal of training stage of ANNs (L→0) [33]. In this research, cross-entropy loss is used. Cross-entropy measures the difference between two probability distributions for a random variable [27,44]. Entropy can be defined as the number of bits needed to represent the information of an event occurring. Lower probability events will have higher entropy, as more information needs to be conveyed, while higher probability events will have low entropy. For a random variable *z*, which can be in a set of discrete states *Z*, with the number of these states being |Z|, with the probability of each of these states occurring being defined as P(z), entropy H(z) can be defined as [45]:(10)$$H\left(Z\right)=\sum_{j=0}^{\left|Z\right|}P\left(z_{j}\right)*log\left(P\left(z_{j}\right)\right).$$

Cross-entropy on the other hand calculates the number of bits required to represent event *z* from distribution P(Z) in comparison to distribution Q(Z). In other words—how many bits are necessary to represent an event using distribution Q(Z), instead of distribution P(Z). When used as a loss function the discrete states *Z* represent the possible classes. Cross-entropy is derived from the Equation (Equation 10), being given as [44]:(11)$$H\left(Z\right)=\sum_{j=0}^{\left|Z\right|}P\left(z_{j}\right)*log\left(Q\left(z_{j}\right)\right).$$

The classes of discriminator output need to be defined. The discriminator is a binary classifier, meaning its’ output will be one of the two classes “0” or “1”. In the presented case the discriminator will return “1” if it classifies the images as “real” and “0” if it classifies the images as fake. To calculate the cross-entropy of generator the output is compared using cross-entropy to an array of ones, because if the generator is performing good, the output of discriminator will be ones for images—as it will classify them as real For the discriminator the real cross-entropy is calculated using discriminator output of real images in comparison to image array of ones (where a value of “1” denotes the class of real image), and fake cross-entropy is calculated in comparison of discriminator output of fake images (expected “0”) to an array of zeros. Following this, cross-entropy of discriminator $H_{d}\left(z\right)$ is calculated as the sum of cross entropies of the real (collected) output $H_{d}^{real}\left(z\right)$ and fake (generated) output $H_{d}^{fake}\left(z\right)$:(12)$$H_{d}\left(z\right)=H_{d}^{real}\left(z\right)+H_{d}^{fake}\left(z\right).$$

The images are generated using a generator and discriminator with the above-described hyperparameters, with the only variation being the number of epochs (iterations) for which the generator is trained to generate new images. Three different epoch ranges are used: 100 epochs, 250 epochs, 500 epochs and 1000 epochs. The number of epochs is an important hyperparameter for GAN performance tuning. Having a number set too low will cause the generator network to be undertrained and images will not be of satisfactory quality. Setting the number of epochs too high will have a negative effect on training times [46]. Still, under training can generate weak results, as it can be seen in Figure 5, where image generated using 100 epochs (Figure 5a) contains lower amount of data. The same figure shows an increase in details from the image obtained while training with 500 epochs (Figure 5c) to image obtained from training the generator over 1000 epochs (Figure 5d). The artifacts apparent in Figure 5c,d are to be expected as a result of previously described transposed convolution process, known as “chekerboard” artifacts [47]. Still, while the effect of number of epochs is visible to the naked eye, the way the number of epochs affects actual data contained within the image in terms of classification scoring cannot be determined in this way, and training and testing need to be performed using predefined CNN architectures, results of which are presented in the following section. It can be possible that images which, to a human observer, appear to be weaker could contain enough data to better the classifier performance, while the images which appear better could be a result of overfitting and fail to do the same [27].

## 4. Description of used CNN Models

With aim of investigating the impact of GAN-based image data augmentation on the urinary bladder cancer diagnosis system, the aforementioned was realized using widely known CNN architectures: AlexNet and VGG16. In this case, the only number of epochs, batch size, solver and their parameters are varied with aim of achieving higher classification performances. A brief description of these CNN models is given in following sections.

### 4.1. AlexNet

AlexNet is a CNN architecture presented in [48]. This architecture was a winner of the 2012 ImageNet Large Scale Visual Recognition Challenge. AlexNet, with its deeper architecture, has fueled the “go deeper” trend that is present in most state-of-the-art computer vision and image classification solutions. It is characterized by nine layer-based architectures whose configuration, slightly adapted to grayscale images, is presented in Table 1.

### 4.2. VGG 16

VGG16 is CNN architecture presented in [49]. Such an approach has offered a significant improvement in comparison with its predecessor AlexNet and it has been proposed at ImageNet Large Scale Visual Recognition Challenge in 2013. This architecture is based on a 16-layered configuration that presents a variation of AlexNet where larger kernels in the first and second convolutional layers are replaced with multiple 3×3 layers. A slightly modified version of VGG16 adapted for grayscale images is presented in Table 2.

Both presented CNN architectures will be trained and tested by using multiple dataset variations that are presented in the following section.

## 5. Dataset Construction Methodology

As stated in previous sections of the article, the collected dataset consists of four classes of urinary bladder mucosa tissue. An overview of dataset characteristics is given in Table 3 where the number of images contained in each class is presented.

The initial dataset will be divided into two parts, one used for classifier training and data augmentation and the other used for testing. The data division procedure will be performed in five different variants, to respect the logic of five-fold cross-validation, as presented in Figure 6. In other words, four equal parts of the dataset will be used for the generation of new images. These images, combined with their predecessors will for the training dataset. The remaining part of the dataset will be used for classifier testing. This procedure will be repeated five times, with the requirement that the remaining fragment of the dataset must differ from previous testing sets.

It is necessary to notice that data used for the generation of new images will not be used during classifier testing. These images will only be combined with generated images in order to form a training dataset. This dataset will be constructed in four different ways (one without generated images and three with different numbers of generated images) in order to examine the influence of the share of generated images in the training dataset on classification performances of proposed CNNs. The number of images in each of the training datasets variations is presented in Table 4.

## 6. Performance Evaluation Measures

Classification performances of AlexNet and VGG16 trained with different configurations of training dataset will be evaluated using multi-class receiver operating characteristic (ROC) analysis. Such an analysis represents a standard evaluation method for binary classifiers [50]. ROC analysis is based on a graphical method of constructing a curve whose *x* coordinate represents the rate of correctly classified positive samples and the *y* coordinate represents the rate of incorrectly classified positive samples. The ROC curve, together with the abscissa and the ordinate encloses the surface with. The area of this surface is called the area under the ROC curve (AUC) and represents a quantified one-dimensional measure of classifier performance. With aim of applying AUC on multi-class classification problem, as is the problem presented in this paper, some modifications of ROC analysis must be performed [21]. Four class classification can be visually interpreted with a confusion matrix:(13)$$M=\left[\begin{array}{cccc} M_{11} & M_{12} & M_{13} & M_{14}\\ M_{21} & M_{22} & M_{23} & M_{24}\\ M_{31} & M_{32} & M_{33} & M_{34}\\ M_{41} & M_{42} & M_{43} & M_{44} \end{array}\right],$$
where elements od the main diagonal represent ratios of correct classifications in a particular class. All other elements represent ratios of some form of incorrect classifications. As stated above, ROC analysis is based on a graphical representation of the relationship between correctly classified positive samples and incorrectly classified positive samples. Such an approach is not straight-forward in the case of multi-class classification. For these reasons, micro-average AUC ($AUC_{micro}$) and macro-average AUC ($AUC_{macro}$) are introduced. A brief descriptions of $AUC_{micro}$ and $AUC_{macro}$ are presented in the following paragraphs.

### 6.1. Micro-Average AUC

Micro-average AUC, as it is in the case of binary ROC analysis, can be expressed by using true positive rate and false positive rate (TPR and FPR). For the case of binary ROC analysis, TPR can be defined as:(14)$$TPR=\frac{P_{C}}{P_{C}+P_{I}},$$
where $P_{C}$ represents a number of correct classifications in positive class and $P_{I}$ represents a number of incorrect classification into positive class. In the case of micro-average AUC, these rates are also calculated by using all confusion matrix elements. By following the stated logic, $TPR_{micro}$ is defined as:(15)$$TPR_{micro}=\frac{tr(M)}{G(M)},$$
where tr(M) represents the trace of the confusion matrix defined as:(16)$$tr\left(M\right)=\sum_{m=1}^{N}M_{mm},$$
and $G_{M}$ represents the sum of all confusion matrix elements:(17)$$G\left(M\right)=\sum_{m=1}^{N}\sum_{n=1}^{N}M_{mn}.$$

On the other hand, $FPR_{micro}$ is defined as:(18)$$FPR_{micro}=\frac{G(M)-tr(M)}{G(M)}.$$

### 6.2. Macro-Average AUC

For the case of macro-average AUC, TPR is expressed as an average of individual TPR values, expressed for each class separately. Such average value can be defined as:(19)$$TPR_{macro}=\frac{1}{N}\sum_{n=1}^{N}TPR_{n}.$$

An example of individual TPR value can be given for the fist class as:(20)$$TPR_{1}=\frac{M_{11}}{M_{11}+M_{12}+M_{13}+M_{14}}.$$

Analogously, $FPR_{macro}$ can be defined as:(21)$$FPR_{macro}=\frac{1}{N}\sum_{n=1}^{N}FPR_{n}.$$

An example of individual FPR value is given for the first class and it is defined as:(22)$$FPR_{1}=\frac{M_{21}+M_{31}+M_{41}}{G\left(M\right)-\left(M_{11}+M_{21}+M_{31}+M_{41}\right)}.$$

Presented measures will be applied for each of the five cases of cross-validation. As a final measures, mean $AUC_{micro}$ and mean $AUC_{macro}$ will be used. These mean values will be presented alongside their standard deviations, in order to determine generalization performances of CNNs trained with different variations of the augmented dataset.

### 6.3. Model Quality Estimation

In order to estimate the model with the highest classification performances, all trained models must be compared. For this reason, results achieved with each model are presented as a set of tuples:(23)$$S=\{S_{1},S_{2},...,S_{N}\},$$
where each element represents a tuple of results obtained by each model defined as:(24)$$S_{n}=\left(\bar{AUC_{micro}},\bar{AUC_{micro}},\sigma \left(AUC_{micro}\right),\sigma \left(AUC_{macro}\right)\right).$$

The set *S* is sorted in such a manner that:(25)$$\pi_{1}\left(S_{1}\right)\geq \pi_{1}\left(S_{2}\right)\geq ...\geq \pi_{1}\left(S_{N}\right),$$
where $\pi_{1}$ represent the first element in a tuple, mainly $\bar{AUC_{micro}}$. In the case when:(26)$$\pi_{1}\left(S_{n-1}\right)=\pi_{1}\left(S_{n}\right),$$
two tuples are sorted in such a manner that:(27)$$\pi_{3}\left(S_{n-1}\right)\leq \pi_{3}\left(S_{n}\right).$$

In this case, $\pi_{3}$ corresponds to the value of the third element of a tuple, mainly $\sigma (AUC_{micro})$.

Presented measures will be used for the determination and comparison of the best-achieved results. Furthermore, in order to determine the general performances of entire variation and network architecture, mean, median and standard deviation of $\bar{AUC_{micro}}$ and $\sigma (AUC_{micro})$ will be measured on the population of all results achieved during a grid-search procedure.

## 7. Results and Discussion

In order to compare the achieved results and to determine the influence of GAN-based augmentation on the proposed classifiers, first the performances achieved by classifiers trained with the original dataset must be measured. In this case, it can be noticed that higher results (up to 0.97) are achieved if VGG-16 is used. Furthermore, it can be noticed that slightly lower classification performances are achieved when AlexNet is used for classification, as presented in Table 5.

It can be noticed that the AlexNet architecture that has achieved the highest classification performances is characterized with higher $\sigma (AUC_{micro})$ and $\sigma (AUC_{macro})$. For these reasons it can be concluded that AlexNet has lower generalization ability in comparison with VGG-16. Furthermore, it can be concluded that VGG-16 deals better with the dataset diversity. This property can be attributed to its deeper architecture.

When the influence of GAN-parameters and share of generated images in the training dataset on the performances of the top-ranked AlexNet models are observed, it is interesting to notice that the highest performances are achieved by increasing the number of GAN epochs to the particular level. It is interesting to notice that increasing the share of generated images in the training dataset does not increase classification performances, rather it decreases them. It can be noticed that the lowest classification performances are achieved when the set augmented with images generated by GAN in 100 epochs. However, these classification performances are still higher than classification performances achieved without data augmentation, pointing to the conclusion that GAN-based augmentation has, in general, a positive impact on the image classification performed by using AlexNet. All statement could be supported by data presented in Table 6, where the first column corresponds to nomenclature presented in Table 4.

When results achieved with VGG-16 are compared, it can be noticed that in all cases, $\bar{AUC_{micro}}$ and $\bar{AUC_{macro}}$ up to 0.99 are achieved. As in the case of AlexNet, lower classification performances are achieved if the original training dataset is augmented with images generated by GAN in 1000 epochs. Furthermore, a slight performance decay can be seen in the case when 18180 generated images are used for augmenting the original set. For these reasons, it can be concluded that a higher share of augmented images reduces the performances of top-ranked VGG-16 architectures. All presented facts could be seen in Table 7.

If results achieved with each variation of AlexNet trained with images generated by GAN in 100 consecutive epochs are compared, it can be noticed that the highest median and average $\bar{AUC_{micro}}$ is achieved if the training dataset is augmented with 10,100 GAN-generated images. Furthermore, it can be noticed that the standard deviation of $\bar{AUC_{micro}}$ is showing a significant decreasing trend suggesting the more stable network behavior, as presented in Figure 7a. When $\sigma (AUC_{micro})$ is observed, it can be noticed that lower median and average values were achieved if augmented datasets are used for training. It can be noticed that the lowest median and $\sigma (AUC_{micro})$ values are achieved if 10,100 generated images are added to the training set. The standard deviation of $\sigma (AUC_{micro})$ is also the lowest in the same case, as presented in Figure 7b. Presented results are suggesting improvement from a standpoint of generalization when an augmented training dataset is used. When presented results are summed up, it can be concluded that, in the case of 100 consecutive GAN epochs, the highest performances are achieved if 10,100 images are added to the training dataset.

Similar results are achieved when images are generated with GAN trained for 250 epochs. In this case, a set assembled with original images combined with 10,100 generated images again achieves the highest median and average values of $\bar{AUC_{micro}}$. The difference lays in the fact that the standard deviation of $\bar{AUC_{micro}}$ is the lowest when 2020 generated images are used for training dataset construction, as presented in Figure 8a. When $\sigma (AUC_{micro})$ is observed, it can be noticed that the lowest average value is achieved in the case of 2020 images, while the lowest median value is achieved in the case of 10,100 images. The standard deviation of $\sigma (AUC_{micro})$ achieves the minimum value of 0.026 when Case 2 is observed. By combining results from both $\bar{AUC_{micro}}$ and $\sigma (AUC_{micro})$ standpoint it can be concluded that, again, the best results will be achieved if 10,100 generated images are added to the training dataset. This conclusion could be drawn from the fact that average and median $\bar{AUC_{micro}}$ values are significantly lower in Case 2, as seen in Figure 8b. This fact, due to classification capability, plays the dominant role in variation rankings, regardless of better generalization performances of the configuration presented in Case 2.

When images generated by using GAN executed for 500 consecutive epochs are added to the training dataset, similar results are achieved. In this case, average $\bar{AUC_{micro}}$ values higher up to 0.95 are achieved. Presented value is achieved when configuration marked as Case 3 is used. Is it interesting to notice that median values are higher in Case 2 and Case 4 than in Case 3, as presented in Figure 9a. When the standard deviation of $\bar{AUC_{micro}}$ is observed, it can be seen that the lowest value is achieved in Case 3. For these reasons, it can be concluded that the presented configuration has the best consistency of classification performances, regardless of batch size, the number of training epochs and solver used. When generalization performances are observed, it can be noticed that the lowest average $\sigma (AUC_{micro})$ is achieved in Case 3. Furthermore, it is interesting to notice that the median value is lower in Case 2. When the standard deviation of all achieved $\sigma (AUC_{micro})$ values is observed, it can be noticed that the lowest standard deviation is also achieved in the case when 10,100 generated images are added to the training set, as presented in Figure 8b. From presented results, it can be concluded that the best performances are achieved when Case 3 is used for the construction of the training dataset. This property is valid from both classification and generalization standpoints.

When the training dataset is augmented with images generated by GAN in 1000 consecutive epochs, slightly different conclusions could be drawn. In this case, a decay of median and average values of $\bar{AUC_{micro}}$ can be noticed, as presented in Figure 10a. It can be noticed that by increasing the share of generated data in the training dataset, no significant change in median and average value can be noticed. Furthermore, a significant fall of standard deviation can be noticed, pointing towards the conclusion that by using a lager share of generated images, a significantly lower change in classification performances is achieved. When generalization performances are observed, it can be noticed that the lowest median and average values of $\sigma (AUC_{micro})$ are achieved in the case when 2020 generated images are used for testing dataset augmentation. Furthermore, it can be noticed that the lowest standard deviation is also achieved in this case, as presented in Figure 10b. From presented results, it can be noticed that the best results are, in this case, achieved when 2020 generated images are used for data augmentation. When results obtained for all four GAN configurations are summed up, it can be concluded that the dominantly highest performances, both from classification and generalization standpoints, are achieved when images are generated by using GAN for 500 consecutive epochs.

When the results achieved with VGG-16 architecture trained with augmented set generated by GAN executed for 100 executive epochs are observed, it can be noticed that a significant improvement of classification performances can be seen only if 10,100 generated images are added to the training dataset. In all other cases, a significant increase can be seen only when median values are observed, as presented in Figure 11a. Furthermore, a significant increase in the standard deviation of $\bar{AUC_{micro}}$ can be noticed, suggesting more unstable behavior of networks trained with augmented datasets. When generalization performances are observed, it can be noticed that if augmented training sets are used, higher average values of $\sigma (AUC_{micro})$ are achieved. On the other hand, a significant decrease in median values can be noticed for Case 3 and Case 4. The standard deviation of $\sigma (AUC_{micro})$ does not show a significant difference, regardless of the share of generated images, as presented in Figure 11b. Presented results are suggesting that there is no significant positive impact on generalization performances of VGG-16 when it is trained by using the training dataset augmented with images generated by GAN in 100 epochs.

When images added to the training dataset are generated by GAN in 250 epochs, similar trends could be noticed. In this case, a slight increase in VGG-16 performances could be noticed. It is interesting to notice that higher median values are achieved when the training dataset is augmented by using 10,100 and 18,180 augmented images. On the other hand, there is no significant change of average $\bar{AUC_{micro}}$ values, as presented in Figure 12a. By observing the change of standard deviation of $\bar{AUC_{micro}}$. When generalization performances are observed, it can be noticed a significant decrease of median $\sigma (AUC_{micro})$ can be noticed, as presented in Figure 12b. On the other hand, there significant decrease of average $\sigma (AUC_{micro})$ and its standard deviation, which is even higher. These results are pointing towards the conclusion that there is no increase in generalization performances.

When classification performances of VGG-16 trained with augmented dataset constructed with images generated by GAN in 500 epochs are compared, a slight increase of average and median value of $\bar{AUC_{micro}}$ can be noticed. In the same time, due to higher standard deviations, a conclusion that defines a more unstable behavior of VGG-16 networks trained with augmented dataset can be drawn. All presented properties are shown in Figure 13a. When generalization performances are observed, it can be noticed that there is no significant decrease of median $\sigma (AUC_{micro})$. On the other hand, a decrease in median value can be noticed, as presented in Figure 13b. As it is in the case of all presented variations of training dataset used for training of VGG-16, there is no significant change in generalization capabilities of such a network. On the other hand, only limited improvements are achieved in classification performances.

As the last case, VGG-16 trained with a dataset augmented with images generated by GAN in 1000 consecutive epochs. When classification performances are compared, a significant increase of the median value of $\bar{AUC_{micro}}$ can be noticed in Case 2 and Case 3. On the other hand, there is no significant increase in average value, as presented in Figure 14a. Furthermore, it can be noticed that the standard deviation of $\bar{AUC_{micro}}$ is significantly larger if augmented datasets are used, suggesting the less stable behavior. When generalization performances are observed, it can be noticed that by increasing the share of GAN-generated images, lover median values of $\sigma (AUC_{micro})$ are achieved. This property is valid only for Cases 3 and 4. On the other hand, there is no decrease in average value. In other words, average $\sigma (AUC_{micro})$ is lower when the original dataset is used for the training of VGG-16. When the standard deviation of $\sigma (AUC_{micro})$ is observed, no significant change can be noticed, as presented in Figure 14b. From presented facts, it can be concluded that there is an increase in generalization performances of VGG-16 when images generated in 1000 epochs are used for training dataset augmentation.

When all obtained results are compared, it can be noticed that by using AlexNet trained with augmented dataset constructed with 10,100 images generated with 500 GAN epochs significantly larger average and median values of $\bar{AUC_{micro}}$ are achieved. Furthermore, the most stable performance is also achieved by using this configuration, judging by the lowest value of the standard deviation. Furthermore, if VGG-16 architecture trained with augmented dataset constructed with 18180 images generated trough 250 epochs is used, no significant improvement in regard to the average and median value of $\bar{AUC_{micro}}$ is achieved. Presented values are only slightly higher than values achieved with VGG-16 architecture trained with the original dataset, as presented in Figure 15a. When generalization performances are compared, it can be noticed that by using AlexNet architecture lower average and median values of $\sigma (AUC_{micro})$, as well as its standard deviation. These results are pointing to the conclusion that AlexNet trained with an augmented dataset has achieved higher generalization performances, in comparison with networks trained with the original dataset. On the other hand, it can be noticed that VGG-16 architecture trained with augmented dataset achieves higher average, median and standard deviation of $\sigma (AUC_{micro})$, as presented in Figure 15b. These results are pointing to the conclusion that VGG-16 architecture, trained with augmented dataset performs poorer than networks trained with original dataset.

## 8. Final Remarks and Conclusions

According to the presented results, it can be seen that GAN-based augmentation has an influence on the classification and generalization performances of CNNs. It can be noticed that, in case of AlexNet, such an approach does not increase significantly performances of best configuration but rather it increases average and median performances. Although not significant, the improvement of the best configuration is visible in classification and generalization. For this reasons, it can be concluded that GAN-based augmentation has enabled a significantly more stable network behavior and made it less sensitive to changes in the number of epochs, batch sizes and solvers. Furthermore, a significant improvement can be seen from a standpoint of generalization. Although promising for AlexNet, the results for the VGG-16 show no significant improvement except from the standpoint of the best configuration, whose classification and generalization capabilities have been increased. For this reason, it can be concluded that the improvement depends not only on the GAN configuration and the share of generated images in the training dataset, but also on the network architecture itself. In accordance with the hypotheses set forth in the introduction, the following conclusion can be drawn:It is possible to utilize GAN for augmentation of urinary bladder cancer image dataset,It can be noticed that the classifier performances are higher if an augmented set is used,By increasing the share of generated images, classifier performances are increasing to a certain level and then decreasing.

## Figures and Tables

**Figure 1 biology-10-00175-f001:**
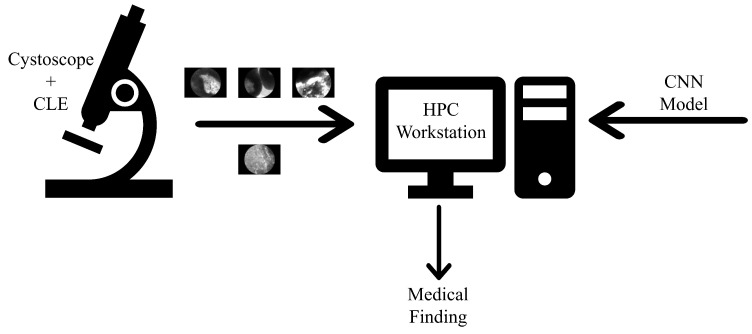
Dataflow diagram of Convolutional Neural Network-based (CNN) utilization in urinary bladder cancer diagnosis.

**Figure 2 biology-10-00175-f002:**
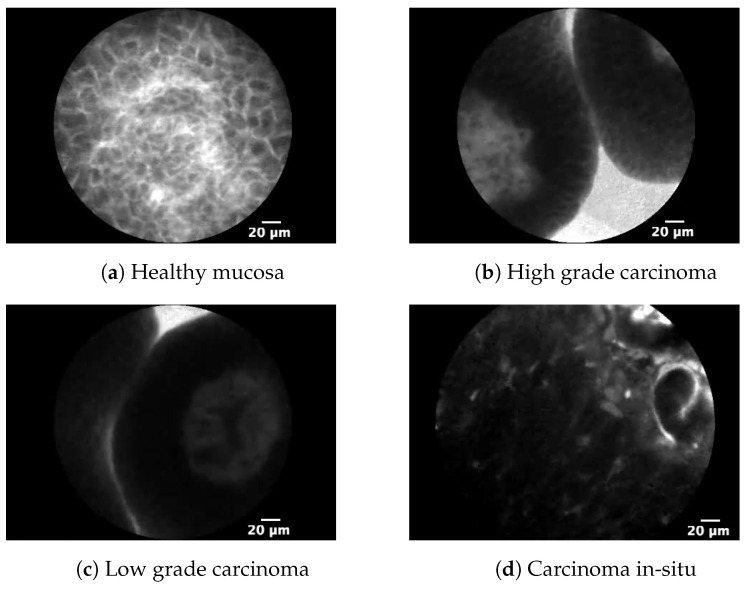
Representation of all four cystoscopy dataset classes.

**Figure 3 biology-10-00175-f003:**
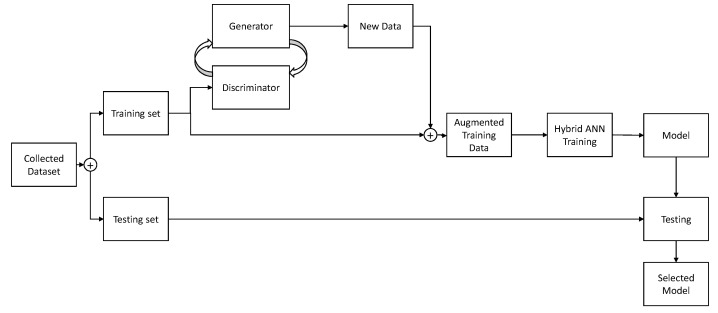
The process of Generative Adversarial Networks (GAN) use for a single image type is shown. Data is split into training and testing sets, and training set is used to generate additional images. Generated images are mixed with existing dataset and used for training, generating models which are finally evaluated on the testing set.

**Figure 4 biology-10-00175-f004:**
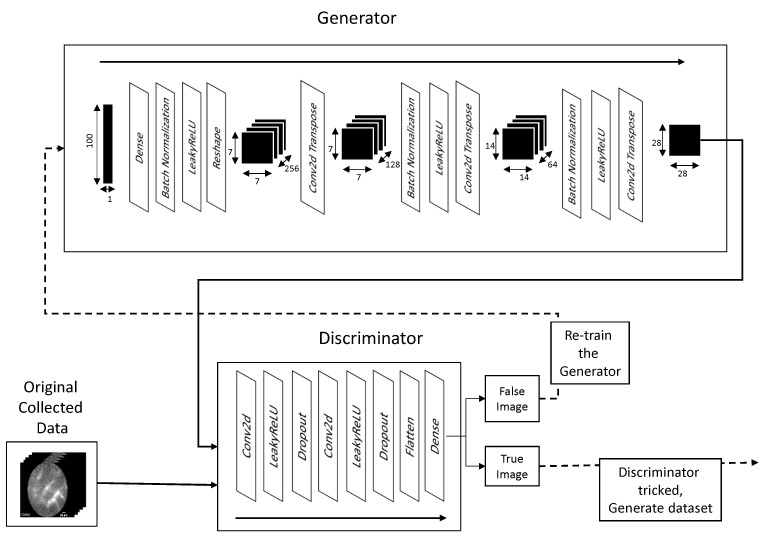
Illustration of generation process of generation and discrimination of images.

**Figure 5 biology-10-00175-f005:**
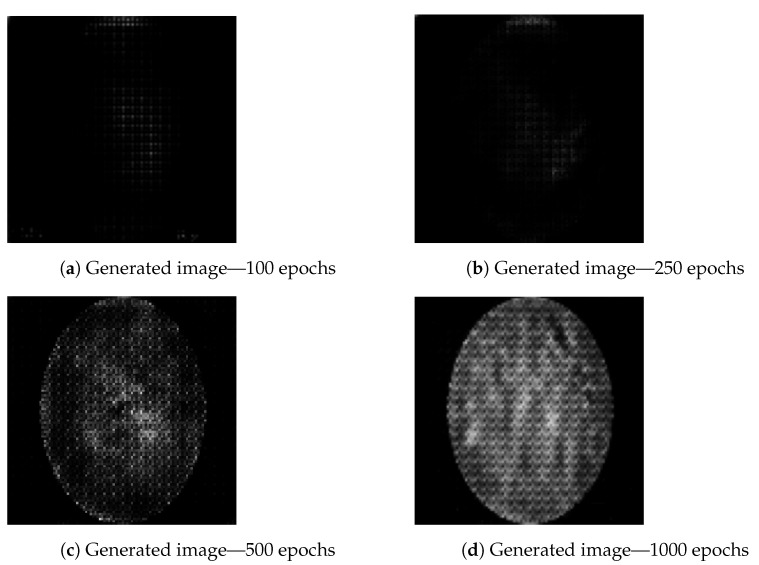
Comparison of images representing healthy mucosa generated by GAN executed for: 100 (**a**), 250 (**b**) 500 (**c**) and 1000 (**d**) epochs respectively.

**Figure 6 biology-10-00175-f006:**
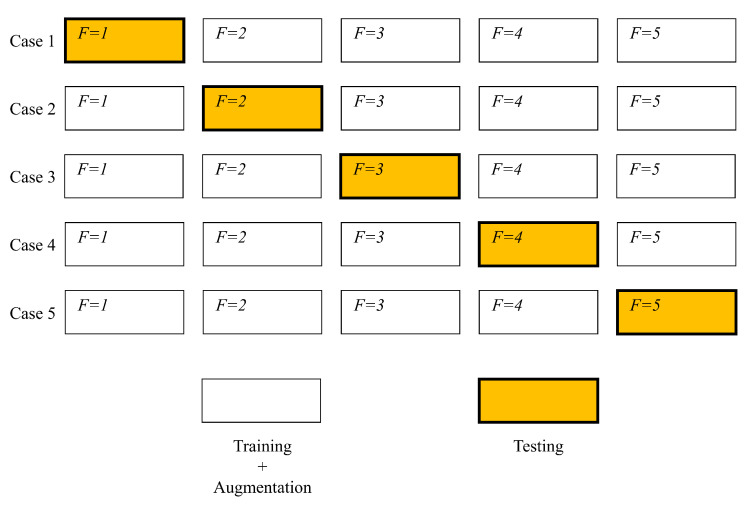
Illustration of a cross-validation procedure adapted for GAN-based augmentation.

**Figure 7 biology-10-00175-f007:**
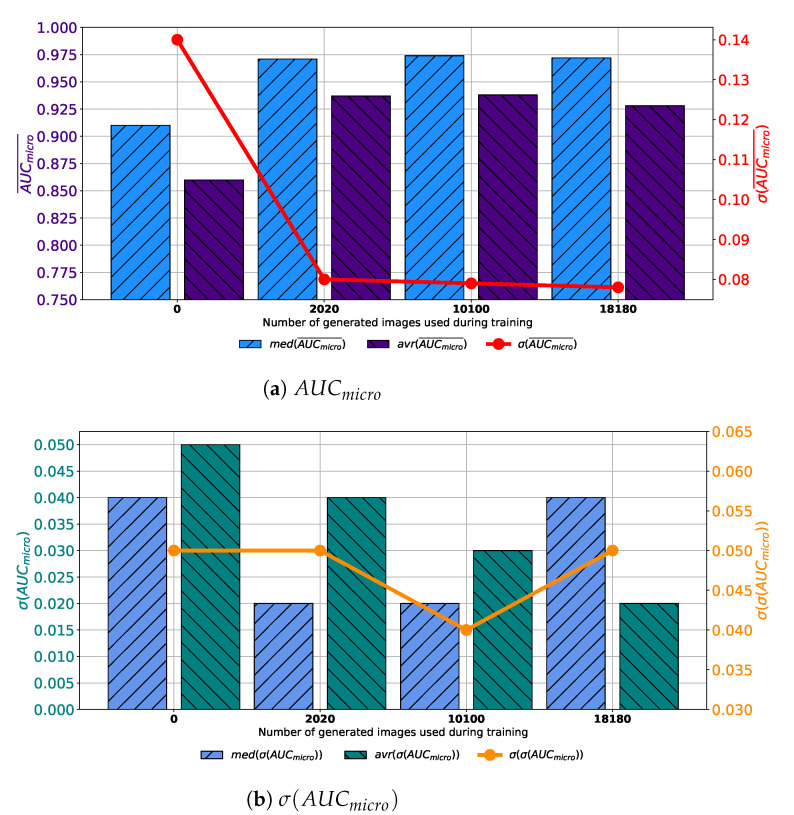
Median, average $\bar{AUC_{micro}}$ and standard deviation of $\bar{AUC_{micro}}$ and $\sigma (AUC_{micro})$ for the case of AlexNet trained with images generated by GAN in 100 epochs.

**Figure 8 biology-10-00175-f008:**
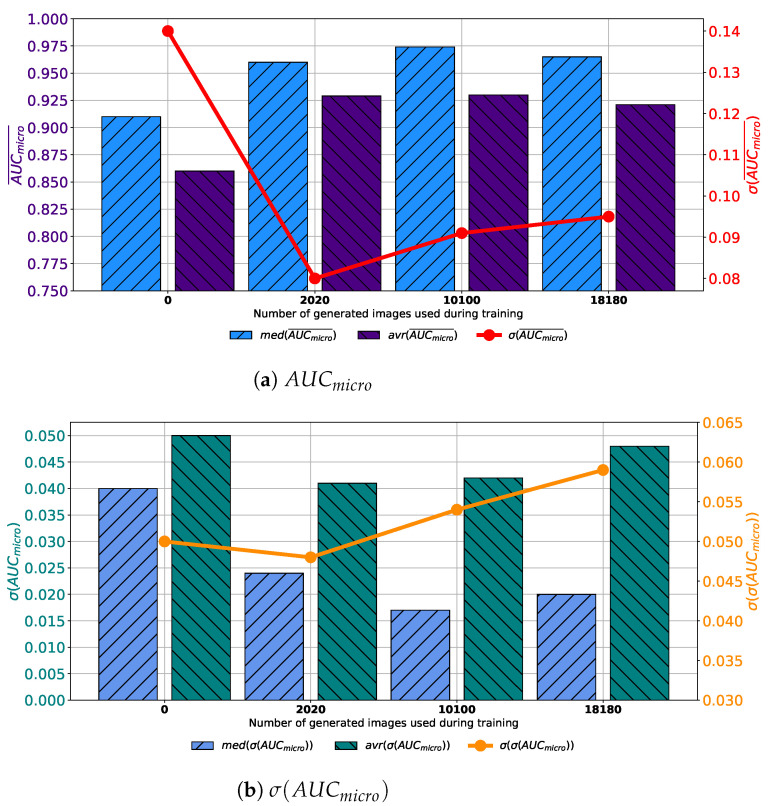
Median, average $\bar{AUC_{micro}}$ and standard deviation of $\bar{AUC_{micro}}$ and $\sigma (AUC_{micro})$ for the case of AlexNet trained with images generated by GAN in 250 epochs.

**Figure 9 biology-10-00175-f009:**
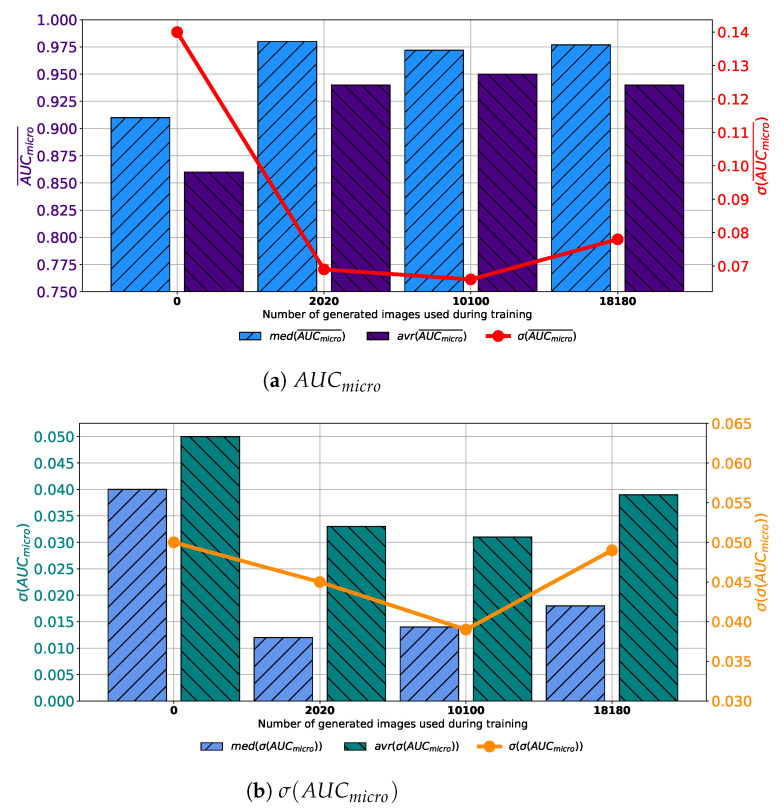
Median, average $\bar{AUC_{micro}}$ and standard deviation of $\bar{AUC_{micro}}$ and $\sigma (AUC_{micro})$ for the case of AlexNet trained with images generated by GAN in 500 epochs.

**Figure 10 biology-10-00175-f010:**
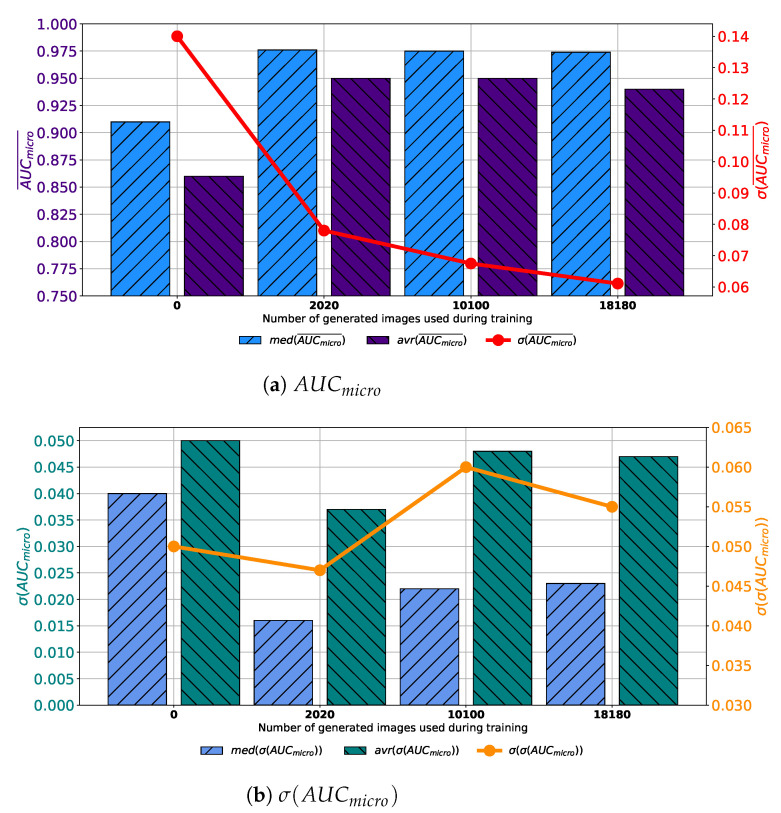
Median, average $\bar{AUC_{micro}}$ and standard deviation of $\bar{AUC_{micro}}$ and $\sigma (AUC_{micro})$ for the case of AlexNet trained with images generated by GAN in 1000 epochs.

**Figure 11 biology-10-00175-f011:**
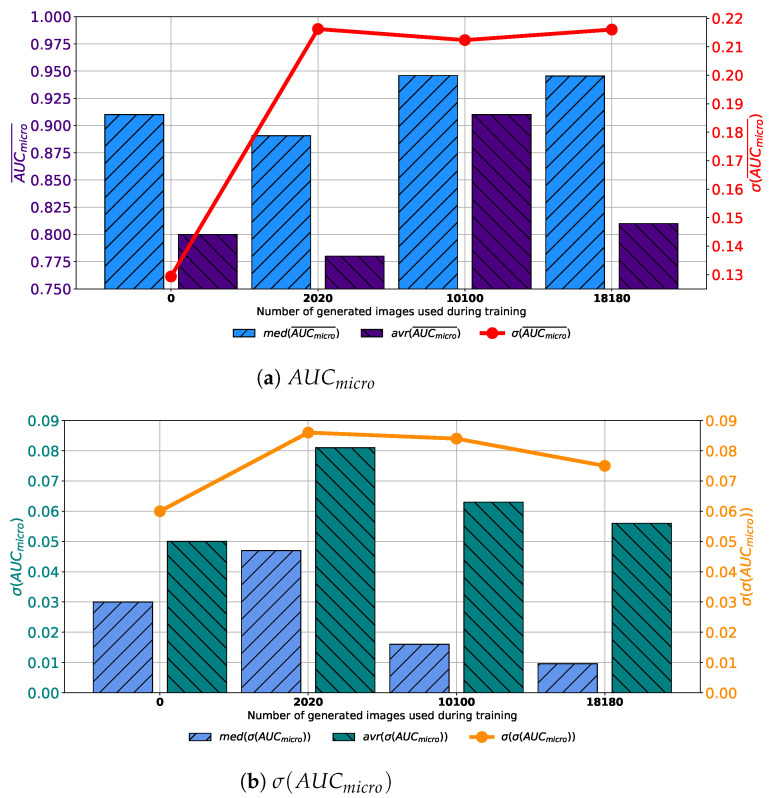
Median, average $\bar{AUC_{micro}}$ and standard deviation of $\bar{AUC_{micro}}$ and $\sigma (AUC_{micro})$ for the case of VGG-16 trained with images generated by GAN in 100 epochs.

**Figure 12 biology-10-00175-f012:**
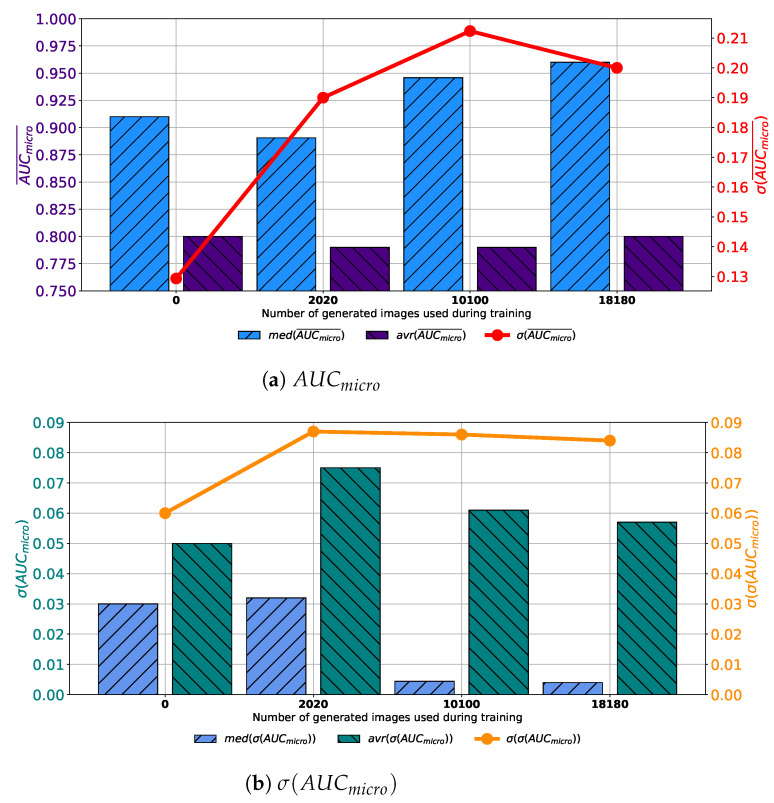
Median, average $\bar{AUC_{micro}}$ and standard deviation of $\bar{AUC_{micro}}$ and $\sigma (AUC_{micro})$ for the case of VGG-16 trained with images generated by GAN in 250 epochs.

**Figure 13 biology-10-00175-f013:**
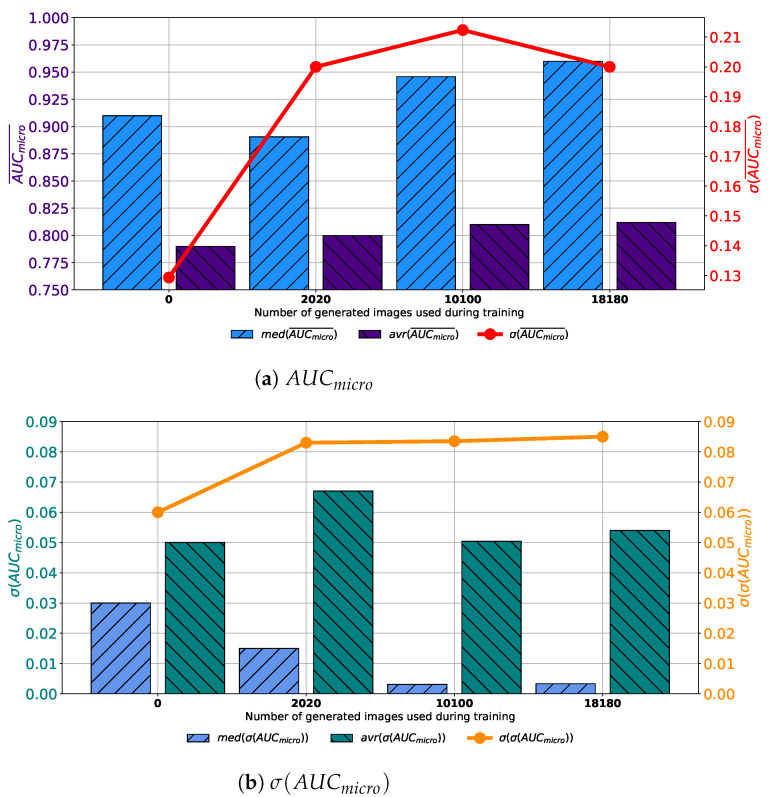
Median, average $\bar{AUC_{micro}}$ and standard deviation of $\bar{AUC_{micro}}$ and $\sigma (AUC_{micro})$ for the case of VGG-16 trained with images generated by GAN in 500 epochs.

**Figure 14 biology-10-00175-f014:**
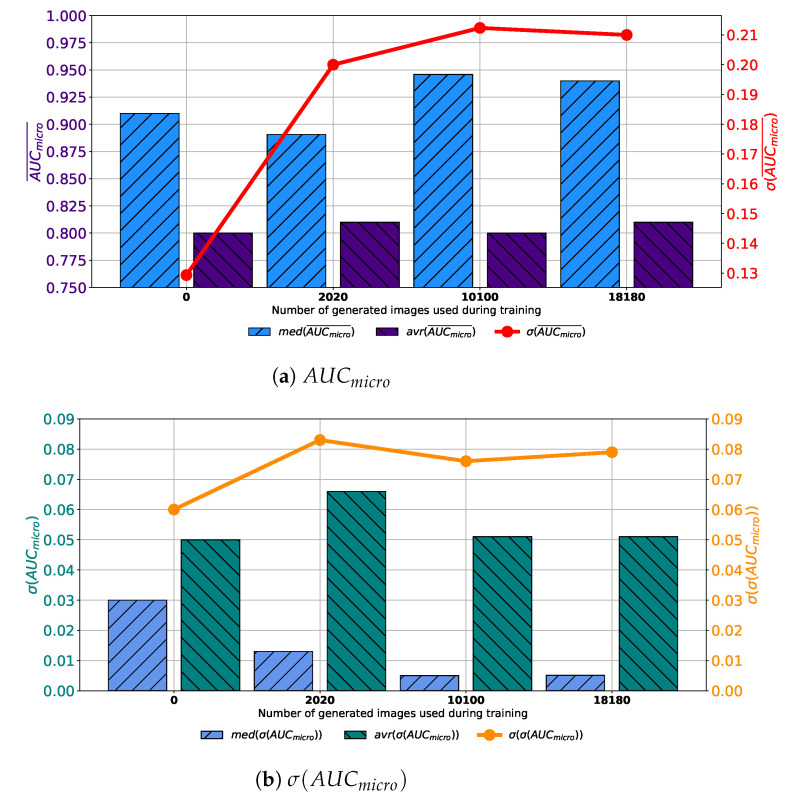
Median, average $\bar{AUC_{micro}}$ and standard deviation of $\bar{AUC_{micro}}$ and $\sigma (AUC_{micro})$ for the case of VGG-16 trained with images generated by GAN in 1000 epochs.

**Figure 15 biology-10-00175-f015:**
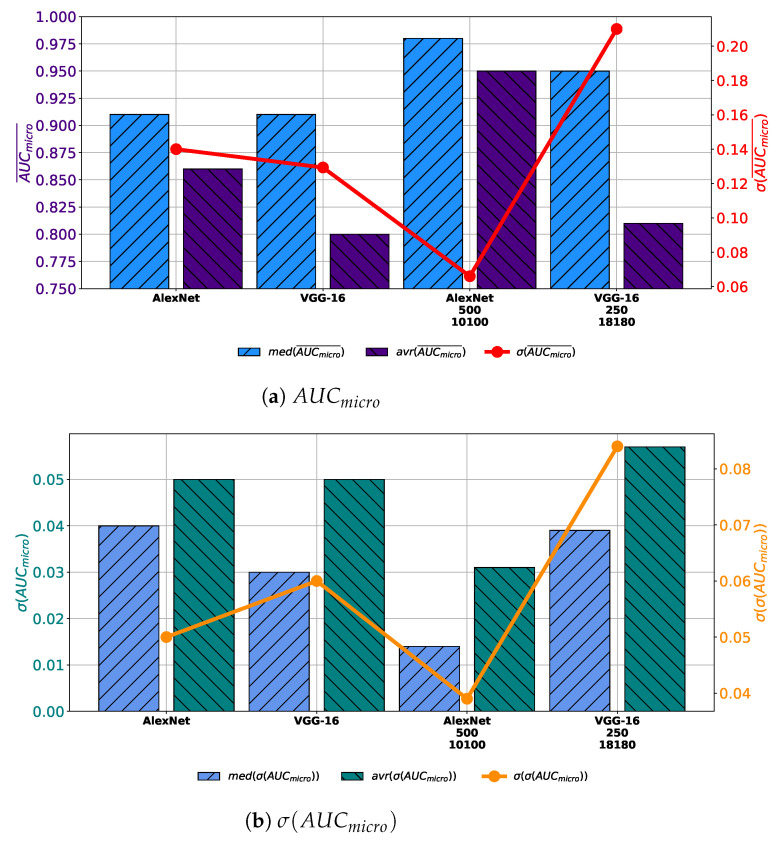
Comparison of achieved results.

**Table 1 biology-10-00175-t001:** Description of AlexNet architecture (C—convolutional layer, P—Max pooling, FC—fully connected).

Layer	Type	Feature Map	Size	Kernel Size	Stride	Activation Function
Input	Image	1	227×227×1	-	-	-
1	C	96	55×55×96	11×11	4	ReLU
	P	96	27×27×96	3×3	2	-
2	C	256	27×27×256	5×5	1	ReLU
	P	256	13×13×256	3×3	2	-
3	C	384	13×13×384	3×3	1	ReLU
4	C	384	13×13×384	3×3	1	ReLU
5	C	256	13×13×256	3×3	1	ReLU
	P	256	6×6×256	3×3	2	-
6	FC	-	9216	-	-	ReLU
7	FC	-	4096	-	-	ReLU
8	FC	-	4096	-	-	ReLU
Output	FC	-	4	-	-	Softmax

**Table 2 biology-10-00175-t002:** Description of VGG 16 architecture (C—convolutional layer; P—Max pooling; FC—fully connected).

Layer	Type	Feature Map	Size	Kernel Size	Stride	Activation Function
Input	Image	1	224×224×1	-	-	-
1	2×C	96	224×224×64	3×3	1	ReLU
	P	64	112×112×64	3×3	2	-
3	2×C	128	112×112×128	3×3	1	ReLU
	P	256	56×56×128	3×3	2	-
5	2×C	256	56×56×256	3×3	1	ReLU
	P	384	28×28×256	3×3	2	ReLU
7	3×C	512	28×28×512	3×3	1	ReLU
	P	256	14×14×512	3×3	2	-
10	3×C	512	14×14×512	3×3	1	ReLU
	P	512	7×7×512	3×3	2	-
13	FC	-	25,088	-	-	ReLU
14	FC	-	4096	-	-	ReLU
15	FC	-	4096	-	-	ReLU
Output	FC	-	4	-	-	Softmax

**Table 3 biology-10-00175-t003:** Number of instances in all four classes used in the presented research.

	Number of Images			
	Non-cancer tissue	High grade cancer	Low grade cancer	CIS
	900	600	680	345

**Table 4 biology-10-00175-t004:** Variations in number of original and generated images used for classifier training.

Case	Number of Original Images	Number of Generated Images	Total Number of Images
1	2020	0	2020
2	2020	2020	4040
3	2020	10,100	12,120
4	2020	18,180	20,200

**Table 5 biology-10-00175-t005:** Results achieved with AlexNet and VGG16 architectures without data augmentation (Case 1).

Architecture	Solver	Batch Size	Number of Epochs	$\bar{{\mathrm{AUC}}_{\mathrm{micro}}}$	$\sigma_{{\mathrm{AUC}}_{\mathrm{macro}}}$	$\bar{{\mathrm{AUC}}_{\mathrm{macro}}}$	$\sigma_{{\mathrm{AUC}}_{\mathrm{macro}}}$
AlexNet	RMSprop	16	10	0.96	0.04	0.96	0.05
VGG16	Adam	16	7	0.97	0.01	0.97	0.01

**Table 6 biology-10-00175-t006:** Results achieved with AlexNet architecture.

Case	GAN Epochs	Solver	Batch Size	Number of Epochs	$\bar{{\mathrm{AUC}}_{\mathrm{micro}}}$	$\sigma_{{\mathrm{AUC}}_{\mathrm{micro}}}$	$\bar{{\mathrm{AUC}}_{\mathrm{macro}}}$	$\sigma_{{\mathrm{AUC}}_{\mathrm{macro}}}$
	100	AdaDelta	32	9	0.98	0.01	0.98	0.013
2	250	AdaMax	32	9	0.99	0.003	0.99	0.003
	500	AdaMax	32	8	0.99	0.003	0.99	0.003
	1000	AdaMax	32	10	0.99	0.003	0.99	0.002
	100	AdaMax	4	9	0.98	0.02	0.98	0.011
3	250	AdaMax	32	8	0.99	0.002	0.99	0.002
	500	AdaMax	32	5	0.99	0.001	0.99	0.001
	1000	AdaMax	32	9	0.99	0.003	0.99	0.003
	100	AdaDelta	8	8	0.98	0.005	0.98	0.005
4	250	AdaGrad	16	10	0.99	0.004	0.99	0.004
	500	AdaMax	32	10	0.99	0.003	0.99	0.003
	1000	AdaMax	32	10	0.99	0.002	0.99	0.002

**Table 7 biology-10-00175-t007:** Best results achieved with VGG-16 architecture.

Case	GAN Epochs	Solver	Batch Size	Number of Epochs	$\bar{{\mathrm{AUC}}_{\mathrm{micro}}}$	$\sigma_{{\mathrm{AUC}}_{\mathrm{macro}}}$	$\bar{{\mathrm{AUC}}_{\mathrm{macro}}}$	$\sigma_{{\mathrm{AUC}}_{\mathrm{macro}}}$
	100	AdaGrad	8	9	0.99	0.004	0.99	0.004
2	250	AdaGrad	8	8	0.99	0.002	0.99	0.002
	500	AdaGrad	4	6	0.99	0.0007	0.99	0.0007
	1000	AdaGrad	16	10	0.99	0.0004	0.99	0.0004
	100	AdaGrad	8	4	0.99	0.004	0.99	0.003
3	250	AdaDelta	4	8	0.99	0.002	0.99	0.002
	500	AdaGrad	16	6	0.99	0.0005	0.99	0.0005
	1000	AdaGrad	8	9	0.99	0.001	0.99	0.001
	100	AdaDelta	8	10	0.99	0.005	0.99	0.004
4	250	AdaGrad	16	10	0.99	0.002	0.99	0.002
	500	AdaDelta	8	10	0.99	0.002	0.99	0.002
	1000	AdaGrad	16	10	0.99	0.001	0.99	0.001

## Data Availability

The data presented in this study are available on request from the corresponding author, if data sharing is approved by ethics committee. The data are not publicly available due to data protection laws and conditions stated by the ethics committee.

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
