# Peer review of "On Urinary Bladder Cancer Diagnosis: Utilization of Deep Convolutional Generative Adversarial Networks for Data Augmentation"

_biology, 2021, doi:10.3390/biology10030175_

Round 1

Reviewer 1 Report

In this manuscript, authors describe a very comprehensive computational analysis- a machine learning strategy- on urinary bladder cancer diagnosis. The manuscript is overall well written. Especially, the authors paid too much attention for the presenting of the methods section which I highly appreciate in this manuscript. From my point of view, the manuscript is acceptable only with some minor corrections:

  • Line 22 and 23: AlexNet is differently written. Please make it consistent
  • Line 61: CNN is first time used, please write it
  • Line 71: AI is first time used
  • Line 81: ANN is first time used
  • Line 94: data set -> change it as dataset
  • Line 129: image 3 -> Image (please control whole manuscript, equations, tables and images must be start with capital letter)
  • In Equation 1: please refer the parameters n and k
  • Line 160: equation 4 -> Equation 4 -> e capital
  • new matrix C(x, h) is performed using equation 3 -> e capital
  • Line 163: The networks in question -> Equation number is missing and e capital
  • Lines 205 and 207: equation -> e capital
  • Line 249: subfigure 5a -> change it as Figure 5a (do same changes please for the following lines as well )

Author Response

Respected reviewer,

please find answers to your comments in the attached PDF.

Kindest regards

Reviewer 2 Report

This is an interesting article aimed at investigating the influence of Generative Adversarial Networks (GAN)-based image data augmentation of CNNs for urinary bladder cancer diagnosis.  The paper is well written and the data well presented. I have only some minor comments:

  • the abstract is too long
  • Figure 2. Representation of all four cystoscopy dataset classes. Please add a scale bar and provide a description of these images in the text.

Author Response

Respected reviewer,

please find the answers to the comments you posed attached.

Kindest regards,

Authors
